# Impact of Climate Variability and Interventions on Malaria Incidence and Forecasting in Burkina Faso

**DOI:** 10.3390/ijerph21111487

**Published:** 2024-11-08

**Authors:** Nafissatou Traoré, Ourohiré Millogo, Ali Sié, Penelope Vounatsou

**Affiliations:** 1Swiss Tropical and Public Health Institute, Kreuzstrasse 2, CH-4123 Allschwil, Switzerland; nafissatou.traore@swisstph.ch; 2University of Basel, Petersplatz 1, CH-4001 Basel, Switzerland; 3Nouna Health Research Centre, National Institute of Public Health, Nouna BP 02, Burkina Faso; ourohire2001@yahoo.fr (O.M.); sieali@yahoo.fr (A.S.); 4Institut de Recherche en Sciences de la Santé, Centre National de Recherche Scientifique et Technologique, Ouagadougou 03 BP 7047, Burkina Faso

**Keywords:** Bayesian modeling, generalized autoregressive moving average models (GARMA), forecasting skill, lag time, wavelet analysis, seasonality

## Abstract

Background: Malaria remains a climate-driven public health issue in Burkina Faso, yet the interactions between climatic factors and malaria interventions across different zones are not well understood. This study estimates time delays in the effects of climatic factors on malaria incidence, develops forecasting models, and assesses their short-term forecasting performance across three distinct climatic zones: the Sahelian zone (hot/arid), the Sudano-Sahelian zone (moderate temperatures/rainfall); and the Sudanian zone (cooler/wet). Methods: Monthly confirmed malaria cases of children under five during the period 2015–2021 were analyzed using Bayesian generalized autoregressive moving average negative binomial models. The predictors included land surface temperature (LST), rainfall, the coverage of insecticide-treated net (ITN) use, and the coverage of artemisinin-based combination therapies (ACTs). Bayesian variable selection was used to identify the time delays between climatic suitability and malaria incidence. Wavelet analysis was conducted to understand better how fluctuations in climatic factors across different time scales and climatic zones affect malaria transmission dynamics. Results: Malaria incidence averaged 9.92 cases per 1000 persons per month from 2015 to 2021, with peak incidences in July and October in the cooler/wet zone and October in the other zones. Periodicities at 6-month and 12-month intervals were identified in malaria incidence and LST and at 12 months for rainfall from 2015 to 2021 in all climatic zones. Varying lag times in the effects of climatic factors were identified across the zones. The highest predictive power was observed at lead times of 3 months in the cooler/wet zone, followed by 2 months in the hot/arid and moderate zones. Forecasting accuracy, measured by the mean absolute percentage error (MAPE), varied across the zones: 28% in the cooler/wet zone, 53% in the moderate zone, and 45% in the hot/arid zone. ITNs were not statistically important in the hot/arid zone, while ACTs were not in the cooler/wet and moderate zones. Conclusions: The interaction between climatic factors and interventions varied across zones, with the best forecasting performance in the cooler/wet zone. Zone-specific intervention planning and model development adjustments are essential for more efficient early-warning systems.

## 1. Background

Malaria is a severe infectious disease caused by parasites of the Plasmodium genus, transmitted by the female Anopheles mosquitoes. In 2022, it was estimated that there were 249 million malaria cases globally [1], resulting in approximately 608,000 deaths across 85 countries [1]. Compared to the figures recorded in 2021, the cases increased by five million [1]. Most of these cases occurred in the WHO African Region [1]. The incidence of malaria exhibited a notable decline, from 81 cases in 2000 to 58 cases per 1000 in 2022 [1,2]. Following a slight increase of 3% in 2020, incidence rates have remained relatively stable over the past three years.

Malaria transmission is significantly affected by climatic variables, particularly temperature, precipitation, and humidity. These factors create suitable conditions for developing malaria vectors and parasites [3,4]. Rainfall creates breeding sites that facilitate egg-laying by female mosquitoes, while high humidity conditions enhance the daily survival rates of adult mosquitoes. Temperature affects many aspects of malaria transmission dynamics, including the development rates and survivorship of vector mosquitoes’ juvenile and adult forms. It has been shown that above 34 °C, the development of Anopheles larvae is inhibited, reducing the survival of adult Anopheles. Low temperatures, therefore, favor the survival of larvae [5].

Additionally, temperature impacts the development rates of malaria parasites. Likewise, humidity affects the survival rate of mosquitoes as they cannot complete their transmission cycle below a relative humidity of 60% [6]. Floods may increase vector-borne diseases such as malaria by expanding the number and the range of vector’ habitats. Variations in climatic conditions influence malaria incidence by affecting its timing and intensity [7,8,9]. Research findings have underscored the potential for changes in relationships between environmental/climatic factors and malaria incidence due to changes in transmission intensity [10]. Investigations in Uganda, Ghana, and Tanzania have identified significant variations in malaria transmission intensity across ecologically distinct sites [11,12,13]. In India, a study highlighted a significant difference in the distribution of malaria incidence across tropical, temperate, polar, and cold climate types [14]. Other studies have reported that delay times (lags) between climatic factors and incidence may vary across regions or ecological settings. For instance, a study in Mali has revealed differences in the relationship between malaria incidence and environmental conditions in lag duration across two different ecological zones [15]. In Ethiopia, rainfall and the minimum temperature were associated with a delayed increase in malaria cases in cold districts, while the associations in the hot districts occurred at shorter lags [16].

Forecasting models play a crucial role in malaria surveillance and decision-making, empowering policymakers and public health professionals to anticipate the future prevalence of the disease and take proactive measures [17]. Several malaria forecasting models have been developed in many endemic countries in Africa, using various statistical methods, mathematical modeling, and machine learning approaches [4,5,7,17,18,19], but few have accounted for regional differences in climate conditions or integrated malaria control interventions, which are critical for accurate predictions. These models have typically used climatic/environmental data such as temperature, rainfall, and vegetation to forecast malaria cases over a specific period. Studies in Ethiopia have shown that forecasting accuracy varies significantly across climatic conditions [20]. Similarly, variations in forecasting performance across different settings were observed in Uganda [21].

Burkina Faso ranks among the top six countries globally for malaria cases and deaths, with its incidence comprising 3.2% of the total in 2022 [1]. *Plasmodium falciparum* accounts for 90% of the cases [22]. In 2020, over 11.3 million people were affected, resulting in nearly 508,097 severe cases and 3966 deaths, with a 92% diagnostic confirmation rate [23]. While the country has implemented various interventions, including artemisinin-based combination therapies (ACTs), insecticide-treated nets (ITNs), indoor residual spraying (IRS), and seasonal malaria chemoprevention (SMC) [24,25], malaria remains highly seasonal, peaking from July to October [26,27]. Furthermore, Burkina Faso’s distinct Sahelian, Sudano-Sahelian, and Sudanian zones exhibit different climatic conditions, complicating health planning [28].

Few studies have been done in Burkina Faso to predict malaria epidemics. For instance, Harvey et al. (2021) used Gaussian processes and random forest regression to estimate the trajectory of malaria cases over 13 weeks at a health facility based on rainfall data [29]. Bationo et al. (2021) developed a spatio-temporal model to predict malaria incidence in the southwest of Burkina Faso, finding optimal predictive performance at 9 and 16 weeks for rainfall and temperature, respectively [30]. Another study applied Bayesian modeling to data from the weekly epidemiological surveillance system for early warning (OWT) of clinical malaria incidence [31]. However, existing models have focused on climatic factors, overlooking the effects of malaria control interventions and the country’s climatic diversity [29]. For instance, ineffective antimalarial treatments have increased cases at health facilities [21], and the OWT reports cases regardless of whether they are parasitologically confirmed or symptom-based [31].

This study addresses the abovementioned gaps by assessing the interactions between climatic factors and malaria interventions across Burkina Faso‘s three distinct climatic zones. We estimated time delays in the effects of climatic factors on malaria incidence, developed forecasting models, and assessed their performance by climatic zone. We applied Bayesian generalized autoregressive moving average (GARMA) models to the confirmed malaria cases in children under five years old and investigated the seasonal incidence pattern and climatic predictors. We used Bayesian variable stochastic selection to identify the best lag times of climatic predictors in the forecasting models. We conducted wavelet analysis to understand better how fluctuations in climatic factors across different time scales and climatic zones affect malaria transmission dynamics. The data were extracted from the District Health Information System (DHSI2), covering 2015 to 2021.

Research questions:What are the time delays between climatic factors and malaria incidence across the three climatic zones of Burkina Faso?How do malaria interventions interact with climatic factors to influence malaria incidence across the zones?How well can malaria incidence be forecasted across these zones using Bayesian models?What periodicities exist in malaria incidence and climatic factors, and how do they vary across the zones?

## 2. Methods

### 2.1. Country Profile

Burkina Faso experiences high temperatures and variable rainfall. Three climate zones divide the country from north to south: the Sahelian zone in the north, which is hot/arid with rainfall less than 600 mm per year (mm/year); the Sudano-Sahelian zone on the savanna plateau (Mossi Plateau); with moderate temperatures and rainfall, ranging from 600 to 900 mm/year; and the more humid Sudanian zone, which is cooler/wet, with rainfall between 900 and 1200 mm/year [32]. Each zone experiences a pronounced wet and dry season, with the wet season lasting for two months in the north and extending to six months in the south. The rainy season begins in the southwest from late March to early April, gradually moving toward the center by May and June and reaching the north by June or early July. The dry season from March to May is influenced by the dry, easterly winds that carry hot air across Burkina Faso [22].

### 2.2. Malaria Incidence and Control Interventions Data

We extracted monthly confirmed malaria cases in children under five years old from Burkina Faso’s District Health Information System 2 (DHIS2) [33], also known as Entrepôt National des Données de la Santé (National Health Data Warehouse), which has managed national health data since 2013. A confirmed malaria case was diagnosed parasitologically via microscopy or rapid diagnostic test (RDT). Additionally, data on malaria interventions, including the monthly proportion of children who used ITNs and the proportion of children with malaria treated by ACTs, were extracted from DHIS2 by region. The data spans from January 2015 to December 2021 for the 13 administrative regions of Burkina. We used 1092 observations in our analysis (i.e., 13 regions × 7 years × 12 months).

### 2.3. Climatic Data

We extracted Day (LSTD) and Night (LSTN) Land Surface Temperature images from the Moderate-Resolution Imaging Spectroradiometer (MODIS) [34] satellite, with a spatial resolution of 1 × 1 km^2^ and a temporal one of every 8 days from 2015 to 2021. Daily rainfall data were extracted from the Climate Hazards Group InfraRed Precipitation with Station data (CHIRPS) [35] at a spatial resolution of 5.6 × 5.6 km^2^ for the same period. We summarized the data from both MODIS and CHIRPS as monthly averages. Furthermore, monthly LSTD and LSTN values were averaged to obtain monthly LST temperature proxies.

### 2.4. Statistical Analysis

For each month within the study period, incidence rates were calculated by dividing the total number of confirmed malaria cases among children under five by the monthly population at risk. The incidence rates were scaled to per 1000 persons.

Seven different lag-time variables were constructed for each climate factor. These variables were derived by averaging values over specific periods. These periods included the current month of reported malaria incidence (lag 0), a month prior (lag 1), two months prior (lag 2), three months prior (lag 3), the current and the previous month combined (lag 01), the current month and the two previous months combined (lag 012), and the current and the three previous months combined (lag 0123). Thus, for each climatic covariate, a matrix of dimension 84 × 7 was created, consisting of the lagged observations corresponding to the monthly malaria data during 2015–2021.

Generalized autoregressive moving average GARMA (p, q) negative binomial models (Benjamin et al., 2003) [36] were fitted to the malaria incidence data with orders p and q representing the autoregressive (AR) and moving average (MA) terms, respectively. We assumed that the number of malaria cases from all health facilities within a climatic zone during 2015–2021 follows a negative binomial distribution. Multiple models with different combinations of p and q were fitted, and their deviance information criterion (DIC) was compared to select the optimal orders. The orders from the model with the lowest DIC were chosen. The predictors in the models included the lagged climatic covariates and the malaria interventions. Predictors were standardized by subtracting their mean and dividing by their standard deviation to allow comparison of their effects. We developed a separate model for each climatic zone within the Bayesian framework [37].

Bayesian variable selection through stochastic search [38] was employed to identify the most appropriate lag for each climatic predictor. The posterior probability of including each lag was computed, and the one with the highest probability was selected. This approach was selected due to its capacity to address uncertainty effectively in predictor selection, allowing the identification of the most relevant climatic factors impacting malaria incidence. It ensures model accuracy, avoiding overfitting, which is particularly important given the variability across the distinct climatic zones.

We fitted the models to the 2015–2020 data, and their predictive performance was evaluated on the 2021 data for predictions 1 to 12 months ahead. Predictive performance for each forecast lead time was assessed by comparing the observed malaria cases against the forecasted ones. In particular, the absolute percentage error (APE) was calculated as the difference between the observed cases and the mean of the posterior predictive distribution, divided by the observed cases, and then multiplied by 100. The forecast lead time with the lowest APE indicates the time with the highest predictive performance. The mean absolute percentage error (MAPE), the residual mean squared error (RMSE), and the R-squared (R^2^) were then used to evaluate forecast accuracy across all lead times for each climate zone. Models were fitted using Markov Chain Monte Carlo (MCMC) simulation in just another Gibbs sampler (JAGS) [39]. We ran two Markov chains of 200,000 iterations, each with an initial burn-in period of 20,000 iterations. Convergence was assessed by visual inspection of density and trace plots and analytically using the Gelman-Rubin diagnostics (Rhat) [40], where values below 1.05 indicate acceptable convergence, ideally approaching 1. Model parameters were summarized by their posterior medians and the corresponding 95% Bayesian Credible Intervals (BCI) expressed as incidence rate ratios (IRR). Incidence rate ratios (IRRs) represent the relative change in malaria incidence associated with a one-unit increase in a predictor variable. An IRR greater than 1 indicates an increase in incidence, whereas an IRR less than 1 indicates a decrease. The effect of a predictor was considered statistically important if its 95% BCI did not contain one.

Wavelet analysis [41] was employed to enhance comprehension of the temporal dynamics of malaria transmission in association with climatic factors and to evaluate the variability of these relationships across different time scales for each climatic zone. Given the seasonal and fluctuating nature of malaria incidence, wavelet analysis is particularly well-suited for identifying patterns in both the time and frequency domains, which traditional time-series methods might fail to discern. The wavelet power spectrum was used to visualize power distribution and identify periodicity across different scales and positions in malaria incidence, rainfall, and LST. The wavelet cross-spectrum quantifies the correlation between two nonstationary time series, i.e., malaria with rainfall or LST. It provides information about the frequency and temporal location that the time series are linearly correlated. The phase difference can estimate delays in the relationship between the two time series. It is expressed in radians, from −π to +π, and displayed as arrow angles. An angle of 0 radians (an arrow pointing right) means the malaria time series and that of a climatic factor are in perfect synchrony (peaking and dipping simultaneously). An angle of π radians (an arrow pointing left) means the two time series are perfectly out of synchrony (when one peaks, the other dips). Angles between 0 and π (or 0 and −π) show varying degrees of lead or lag between the two series. A significance level of 5% was for the statistical significance of the observed patterns. The wavelet analysis was carried out using the waveletComp package in R (version 4.2.1) [42]. Detailed descriptions of the analysis and model formulations are provided in Appendix A.

## 3. Results

### 3.1. Descriptive Analysis

Table 1 describes the average monthly malaria cases per 1000 persons, climatic factors, and interventions for the 13 administrative regions, classified into three climatic zones over seven years from 2015 to 2021. Overall, malaria incidence was 9.92 cases per 1000 persons per month. The average monthly rainfall and temperature were 72.37 mm and 28.50 °C, respectively. The proportion of children infected and treated with ACT was 0.88 per month. The coverage of bednet use was 30% per month during 2015–2021. In all zones, the proportion of infected children sleeping under bednets was lower than 50% per month, with an increasing trend over the study period. Lower malaria incidence was observed in the hot/arid zone. Moderate and higher monthly incidences were noted in the moderate and cooler/wet zones. In particular, the east, center-north, and southwest administrative regions, located in the moderate, hot/arid, and cooler/wet climatic zones, respectively, exhibited the highest malaria incidence.

Figure 1 illustrates the geographical distribution of monthly incidence across the three climatic zones mentioned above. Figure 2 shows the temporal trend of monthly malaria incidence, LST, rainfall, coverage of ACT, and bednet use from 2015 to 2021. It indicates a global decrease in incidence in the three climatic zones marked by annual seasonality each year. The highest incidence from 2015 to 2016 was observed in the hot/arid zone. From 2017 to 2021, the incidence was highest in the cooler/wet zone. There has been a slight increase in LST globally from 2015 to 2021, with the highest temperature being in the hot/arid zone and the lowest in the cooler/wet zone. Rainfall was consistently lowest in the hot/arid zone, except in 2015 and 2017, and remained nearly constant elsewhere. High and moderate rainfall levels were observed in the cooler/wet and moderate zones, respectively, peaking annually across the zones. Figure 2 also illustrates that malaria incidence was low from January to June and high from July to October across the climatic zones. Appendix A displays an increase in malaria cases per 1000 persons/month from 2015 (9.91 cases) to 2016 (12.60 cases), followed by a decrease from 2017 (11.69 cases) to 2019 (6.75 cases) before rising again to 8.85 cases per 1000/month in 2020.

### 3.2. Effects of Lagged Climatic Factors and of Interventions on Malaria Incidence

Table 2 shows the best lag times associated with LST and rainfall as determined by the Bayesian variable stochastic selection (BVSS) applied within each GARMA model and zone. The GARMA (1, 1) model had the smallest DIC value (Table 2), indicating the best fit; therefore, we employed that model for inference. The results suggest that the highest posterior probabilities of inclusion were obtained at lag 1 for LST (0.5) and lag 3 (0.4) for rainfall in the cooler/wet zone. In the moderate zone, the lags with the highest inclusion probabilities were 1 month for LST (0.3) and 2 months for rainfall (0.44). For the hot/arid zone, the best lags were 2 previous months for both LST and rainfall, with the inclusion probabilities of 0.51 and 0.50, respectively (Table 2).

Table 3 presents the posterior estimates and their 95% BCIs of the GARMA time-series models fitted separately to data from each climatic zone. LST had a statistically important negative effect on malaria incidence of somewhat similar magnitude across all climatic zones. The effect of rainfall was positive, indicating that the higher the rainfall, the higher the malaria incidence (Table 3). The magnitude of the effect was greatest in the hot/arid zone, where each unit increase in rainfall was associated with a 51% increase in malaria incidence (IRR: 1.51, 95% BCI: 1.33–1.70). In the moderate zone, the effect is slightly lower, with a 24% increase in incidence for each unit increase in rainfall (IRR: 1.24, 95% BCI: 1.11–1.37). The cooler/wet zone also exhibits a notable increase in malaria incidence, with a 36% rise for an increase in rainfall (IRR: 1.36, 95% BCI: 1.24–1.48).

Bednet use had a statistically important protective effect in the cooler/wet and moderate zones (Table 3). In particular, for each additional unit of bednet coverage, the incidence of malaria is observed to decrease by 19% in the cooler/wet zone (IRR: 0.81, 95% BCI: 0.52–0.95) and by 56% in the moderate zone (IRR: 0.44, 95% BCI: 0.33–0.93). In the hot/arid zone, bednet use was not statistically important (Table 3); however, the effect of ACT was important and protective, with an 11% decrease in malaria incidence for each unit increase in ACT coverage (IRR: 0.89, 95% BCI: 0.79–0.97).

### 3.3. Model Predictive Performance

The results of the overall fitting in each climatic zone are shown in Figure 3. The plots show that the actual number of malaria cases is close to the fitted and the forecasted ones. The summary statistics in Table 4 confirm the high predictive performance of the models. The R-squared values for the observed and fitted cases are 0.78, 0.65, and 0.76 (Table 4) for the cooler/wet, moderate, and hot/arid zones, respectively. The model performed worst in the moderate zone, as demonstrated by the highest average APE across all lead times, which is 53% (Table 4). The highest predictive performance was achieved for the cooler/wet zone, indicated by the lowest average of APE of 28% (Table 4).

The best predictive performance was observed at lead times of 2, 3, and 2 months for the hot/arid, cooler/wet, and moderate zones, respectively (Table 4). Figure 4 compares the number of observed cases, the forecasted estimates, and their 95% BCI. The forecast error is shown on the secondary vertical axis. In the hot/arid regions, underestimates (forecasted < observed) were observed for four lead times. In the moderate zone and cooler/wet zone, the model overestimated (forecasted > observed) for ten lead times (Figure 4).

### 3.4. Wavelet Analysis

Figure 5 displays the wavelet power spectra of malaria cases per 1000, LST, and rainfall across the three climatic zones. In the hot/arid climatic zone, there is a distinct periodicity in malaria incidence, with a 12-month cycle observed from 2015 to 2021 and a shorter 6-month cycle noted between 2015 and 2016, as depicted in Figure 5A. LST shows a significant 12-month periodicity throughout the study period, except between 2017 and 2018, and a 6-month cycle from 2015 to 2021 (Figure 5B). Rainfall patterns also display a 12-month cycle for 2015 to 2021, with a non-significant 6-month cycle during the study period (Figure 5C).

In the moderate zone, a consistent 12-month cycle of malaria incidence is observed over the study period, as shown in the wavelet power plot (Figure 5D). Two statistically significant cycles of LST have been detected: a 12-month cycle from 2015 to 2021 and a 6-month cycle from 2015 to 2016 and 2018–2021 (Figure 5E). Rainfall exhibits a significant 12-month periodicity from 2015 to 2021 and a shorter, less prominent 6-month cycle from 2016 to 2021 (Figure 5F).

In the cooler/wet zone, malaria incidence shows a significant 12-month periodicity from 2015 to 2021 (Figure 5G). Temperature reveals a 12-month cycle throughout the study period (Figure 5H), with a 6-month cycle observed in 2015–2016 and 2019–2021. The wavelet power plot for rainfall indicates a significant 12-month periodicity from 2015 to 2021 (Figure 5I). The average wavelet power over 2015–2021 for all climatic zones is shown in Appendix A.

The wavelet cross-coherence spectra in Figure 6A–F demonstrate a strong and statistically significant relationship between climatic factors and malaria incidence from 2015 to 2021. The left and down arrows in Figure 6A,C,E depict an anti-phase relationship between LST and malaria over 12-month periods across all climatic zones, where an increase in temperature leads to decreased malaria. Moreover, the arrows in Figure 6B,D,F reveal an in-phase association of rainfall with malaria cases, in which an increase in rainfall leads to a rise in malaria incidence during 2015–2021.

In the hot/arid and the moderate zones, the phase differences range from 1.5 to 2 months for the 12 months over the same study period. In the cooler/wet zone, these phases vary from 2 to 3 months during the study period. Similarly, the phases between LST and malaria incidence vary from 1.5 to 2 months in the hot/arid, 1 to 1.5 months in the moderate and the cooler/wet zones for the 6-month seasonal scale. For the 12-month cycle, the phases range from 1.5 to 2 months in the hot zone, 1 to 2 months in the moderate zone, and 1 to 1.5 months in the cooler/wet zone.

A summary of the findings of the different analyses by climatic zone is provided in Table 5.

## 4. Discussion

In this study, we examined the interactions between climatic factors and malaria interventions in three climatic zones in Burkina Faso. We estimated time delays in the effects of climatic factors on malaria incidence and developed Bayesian GARMA models to forecast malaria incidence in children under five. We evaluated the predictive performance of these models at various lead times within each climatic zone and investigated the temporal dynamics of the incidence using wavelet analysis.

Our results revealed that malaria incidence varies across climatic zones in Burkina Faso, with the highest rates observed in the cooler/wet zone. These findings align with previous studies [3,7,43,44]. Sangaré et al. (2022) identified the southwest part of Burkina Faso, characterized by its cooler climate, as having a higher likelihood of malaria transmission [44]. The high incidence in this zone can be attributed to the longer rainy season and abundant rainfall [45], which create favorable conditions for mosquito breeding and parasite development. In contrast, the moderate and hot/arid zones had lower incidence rates, probably due to less favorable climatic conditions for vector survival and reproduction.

Malaria incidence decreased steadily from 2016 to 2019 before rising in 2020, most likely due to disruptions in services caused by the COVID-19 pandemic [2]. The effectiveness of malaria interventions varied across climate zones, revealing complex interactions between climate and control interventions. Artemisinin-based combination therapies (ACTs) were statistically important and negatively associated with malaria incidence in the hot/arid zone but not in the cooler/wet and temperate zones. This differential efficacy may be related to the risk of emergence of antimalarial resistance, which may be reduced in hot regions where malaria transmission intensity is lower [46,47,48]. Similarly, findings from a study conducted in Tanzania showed that the reduction of malaria incidence achieved by ACTs was much higher in areas with low initial transmission [27]. It has been established that an intervention’s effectiveness is related to its coverage and transmission levels. The limited effectiveness of ACTs in high-transmission settings may be due to factors such as existing drug resistance, as highlighted in various studies [49].

Bednets showed statistical importance in moderate and cooler/wet zones, aligning with studies from Kenya, Tanzania, and Papua Guinea that consistently demonstrated reduced malaria incidence and related deaths, particularly among young children using bednets [50,51,52,53]. However, in the hot/arid climatic zone, the effect of bednet use did not appear to be statistically important. This could be due to heat-related discomfort in hot/arid areas, which could affect bednet use, or changes in human behavior, such as staying out late at night, which increases the risk of mosquito bites before bedtime. The netting material is a barrier, hindering mosquitoes from biting individuals inside [54]. Infused with insecticides like pyrethroids, bednets repel mosquitoes and reduce mosquito density and lifespan in communities, extending protection to non-users [55]. Similarly, reports from Ethiopia, Tanzania, and Cambodia demonstrated a significant association between late-night outdoor activity and increased malaria transmission [56,57,58].

Different lags of rainfall and LST were associated with malaria incidence in different climatic zones. Rainfall was positively associated with incidence at a 2-month lag in the moderate and the hot/arid zones and at a 3-month lag in the cooler/wet zone. These findings are consistent with studies from Ethiopia [16] and Eritrea [59], which reported similar lags of 10–12 weeks (2–3 months) in cooler districts and 6–10 weeks in hot districts.

In Burkina Faso, rainfall was reported to have a delayed influence on malaria cases with a 9 to 14-week lag, regardless of the transmission period [60]. Another study observed that 16 weeks were identified as a time lag between rainfall and incidence 29 in districts located in the cooler/wet zone of Burkina Faso.

Additionally, it has been observed in the highlands of Kenya that malaria cases increased by 1.4% to 10.7% per month for every 10-mm increase in monthly rainfall at a lag of 2–3 months [61]. For instance, when there is low rainfall one month before, an increase in rainfall occurring more than one month later mitigates the impact of insufficient rainfall. This creates more breeding habitats for mosquitoes and increases their number, resulting in an increased risk of malaria incidence [62]. The mechanism by which rainfall affects malaria incidence is complex, as it both creates breeding habitats and accelerates mosquito life cycles, increasing the risk of transmission during peak rainy seasons. [63].

LST was important and negatively associated with malaria incidence in all the climatic zones, with shorter lag times observed in the cooler/wet and moderate zones (1 month) compared to the hot/arid zone (2 months). Previous studies align with this finding [43,50,64,65]. This result can be attributed to the influence of high temperatures on mosquito viability and reproductive potential, leading to a reduction in vector population size and its ability to transmit the disease [66,67]. Other studies have shown that a slight temperature increase significantly affects malaria transmission in areas with lower average temperatures than those with higher temperatures [16,68,69]. Such differences may reflect different ecological interactions and vector dynamics in each setting.

The predictive performance of our models exhibited variability across climatic zones, with the highest performance observed in the cooler/wet zone. The models demonstrated enhanced accuracy over shorter lead times (2 months) in the hot/arid and moderate zones relative to the cooler/wet zone (3 months). This variation in forecasting skill across zones is consistent with findings from similar studies in Ethiopia and Uganda [20,21]. The better performance in the cooler/wet zone might be due to the shorter transmission season in the hot/arid zone, resulting from the rapid evaporation and drying up of breeding sites, which shortens the rainfall duration and affects transmission [70]. The duration of the rainy season varies, lasting up to nine, six, and three months, respectively, in the cooler/wet, moderate, and hot/arid climatic zones [71].

Overall, the models over-predicted malaria cases, which might be better than an under-prediction. Models that under-predict are likely to fail in issuing warnings during actual epidemics, whereas models that over-predict can potentially issue false alerts [72].

The wavelet power spectrum analysis revealed that the seasonal patterns of malaria incidence and LST varied from 6 to 12 months. A significant periodicity of 12 months was also detected for rainfall across the climatic zones during the study period. These findings are supported by a similar study conducted in Sri Lanka, which corroborates these results [73,74]. Furthermore, the wavelet cross-coherence spectrum indicated that malaria occurrence and temperature were out of phase, whereas they were in phase with rainfall over the three climatic zones. Other studies reported similar results [73,74,75].

The wavelet analysis identified phase differences representing the lag time between climatic factors and malaria occurrence. These lags varied by climatic zone, over time, and at different periodicities. Another study using wavelet analysis in Africa reported that rainfall anticipates malaria with varying time lags, typically around 2 to 3 months but sometimes shorter [75].

GARMA models assume stationarity [76] of the time series and the association between the climatic factor and malaria. However, the malaria and climatic time-series data exhibit nonstationary behaviors, challenging this assumption [73]. Non-stationarity indicates that the mean, variance, and autocorrelation structure of a time-series change over time [77]. Malaria data are characterized by significant shifts in the magnitude, frequency, and timing of transmission, influenced by climatic factors, human interventions, and vector dynamics [77]. Climatic variables such as temperature, rainfall, and humidity significantly impact the breeding and survival of malaria mosquitoes and parasites. Changes in these climatic factors may lead to non-stationarity behavior in malaria incidence data [77]. Human activities, including land cover change and water management practices, introduce artificial patterns and trends in the global water cycle, disrupting climatic factors’ stationarity and subsequently affecting malaria incidence [77]. Urbanization, deforestation, and agricultural practices can also alter mosquito habitats and human-vector interactions, leading to nonstationary malaria transmission patterns [78]. Furthermore, implementing control interventions, such as bednets, indoor residual spraying, and improved access to healthcare, can also alter the dynamics of malaria transmission over time [79].

In contrast, wavelet analysis can capture changes over time and simultaneously identify the time intervals and frequency bands where two time series are correlated. It provides a more flexible and interpretable approach to analyzing time-frequency data. By decomposing the signal into different frequency components, wavelet analysis can identify small patterns and phase differences that may be missed by the BVSS [74]. Therefore, wavelet analysis alongside Bayesian models can enhance understanding of the dynamic relationships between climatic factors and malaria incidence.

While this study provides valuable insights into the complex interplay between climatic factors and malaria incidence, some limitations warrant discussion. Variability in health-seeking behavior among individuals who may seek care outside their immediate geographic and climatic context can complicate the attribution of health outcomes solely to local climatic influences [20]. This diversity in patients’ origins can introduce complexity in accurately attributing health outcomes solely to the local climate. In addition, data quality is an important concern; although we used confirmed malaria cases from DHIS2, variations in reporting completeness and accuracy across health facilities may introduce bias, particularly in remote areas where reporting may be less consistent. Moreover, monthly data may mask finer temporal variations in malaria transmission, and our regional analysis may mask some local variations. Finally, our model does not account for potentially important factors such as socioeconomic status, migration patterns, and land use change, which may influence both climate and malaria incidence. The assumptions inherent in our GARMA models also warrant caution; while we tested various autoregressive and moving average terms, other temporal dependencies may not have been captured.

Despite these limitations, the findings have important implications for public health policy and practice. The varying effectiveness of malaria control strategies across climatic zones indicates that these strategies should be tailored to local conditions. For instance, alternative vector control methods may be more effective in the hot/arid zone where bednets showed limited impact. The greater efficacy of ACT in the hot/arid zone, where transmission is lower, indicates that this zone could be prioritized for ACT distribution. In the cooler/wet zone, where the incidence is high, interventions such as the distribution of bednets should be prioritized, particularly during peak transmission periods. Furthermore, the observed time lags between rainfall and malaria incidence highlight the potential advantages of early, proactive interventions based on rainfall forecasts. For example, the deployment of bednets, the stockpiling of antimalarial treatments, and the initiation of community awareness campaigns two to three months before the peak transmission period could effectively mitigate outbreaks in areas with predictable seasonal patterns.

Future research directions should focus on expanding our understanding of malaria transmission dynamics and improving forecasting models across distinct climatic zones. Studies should explore the impact of additional environmental factors on malaria transmission, including humidity, land cover change, wind speed, and altitude. These elements likely interact with rainfall and temperature in complex ways to influence mosquito populations and parasite development. Incorporating socioeconomic indicators (e.g., poverty levels and access to healthcare) and demographic data (e.g., population density and age structure) could provide a more nuanced understanding of malaria risk factors. Additionally, analyzing patterns of human mobility and migration, both seasonal and long-term, could offer insights into the spread of malaria across different climatic zones and inform targeted intervention strategies. Furthermore, researchers should consider nonstationary models when forecasting malaria incidence in the long term, given the dynamic nature of malaria transmission and climate change.

## 5. Conclusions

In conclusion, our short-term forecasting models, incorporating climatic and intervention predictors, are more reliable in cooler/wet zones than in the hot/arid and moderate zones. The effect of the lagged climatic factors and interventions on malaria incidence varied among climatic zones. Rainfall exhibited a positive association with malaria incidence at longer lags in cooler/wet zones compared to others. LST was important and negatively associated with malaria incidence, with a shorter delayed effect in cooler/wet and moderate zones. Delay times should be considered when distributing interventions across different zones, particularly for periodic interventions like bednets and seasonal malaria chemoprevention. Ahead-of-time predictions with the best predictive performance were identified in the hot/arid and moderate zones within a lead time of 2 months and in the cooler/wet within 3-month lead times. Lead times are important for early warning to allow timely delivery of interventions.

Wavelet spectrum analysis revealed significant annual cycles in malaria incidence, LST, and rainfall from 2015 to 2021. The exploration of lag times between climatic factors and malaria incidence highlights the importance of adapting forecasting models to reflect the dynamic nature of these interactions. This also suggests that while stationary models are appropriate for short-term forecasts, nonstationary models should be considered for long-term predictions. However, it is crucial to acknowledge the limitations of our study, such as potential data reporting biases and the non-inclusion of socioeconomic factors, migration patterns, and other environmental factors that could influence malaria transmission. This study suggests that it may be beneficial to consider climatic variability in malaria control efforts and the development of early-warning systems. In the context of climate change, implementing adaptive zone-specific forecasting models will be essential for effective malaria management. By continually refining these models to account for changing climatic conditions and intervention impacts, we can enhance our ability to predict and respond to malaria outbreaks, which may reduce the burden of this disease in Burkina Faso and similar settings.

## Figures and Tables

**Figure 1 ijerph-21-01487-f001:**
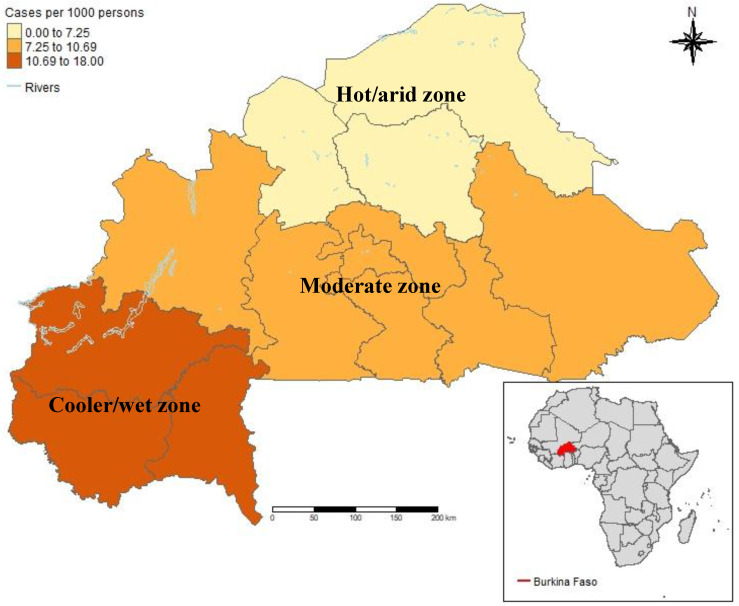
Geographical distribution of monthly average malaria incidence by climatic zone. The range of values in the legend indicates the minimum and maximum values within the zones.

**Figure 2 ijerph-21-01487-f002:**
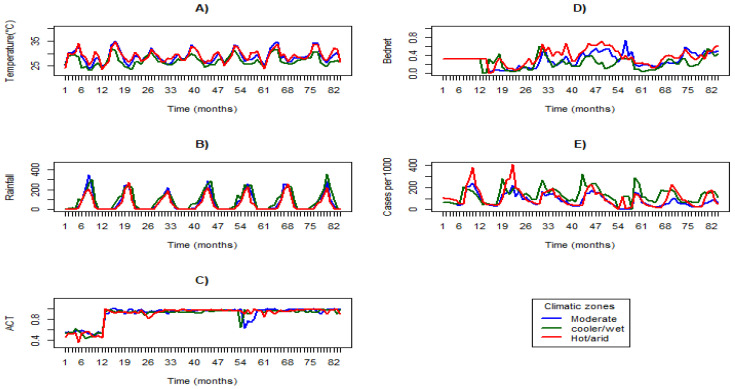
Temporal trend of monthly (**A**) LST, (**B**) rainfall, (**C**) ACT coverage, (**D**) bednet use coverage, and (**E**) malaria incidence.

**Figure 3 ijerph-21-01487-f003:**
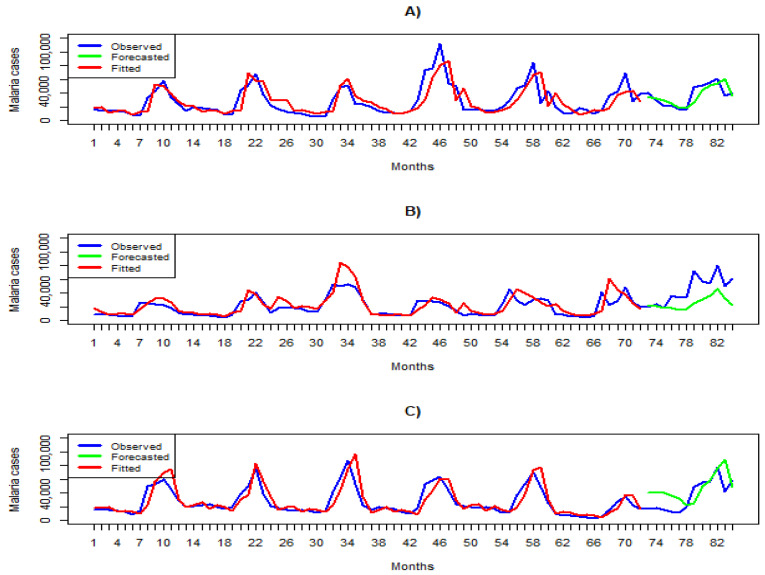
Overall model fitting and predictive performance in the three climatic regions: (**A**) hot/arid zone, (**B**) cooler/wet zone, and (**C**) Moderate zone. The blue, green, and red lines represent actual, forecasted, and fitted cases, respectively.

**Figure 4 ijerph-21-01487-f004:**
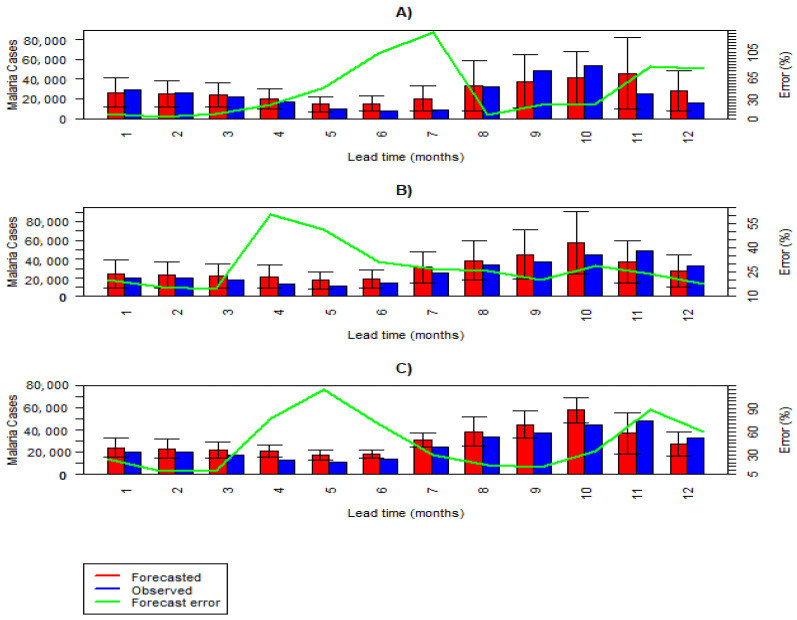
Model predictive performance for each lead time (1 to 12 months) of the forecasting data segment: (**A**) hot/arid zone, (**B**) cooler/wet zone, (**C**) moderate zone.

**Figure 5 ijerph-21-01487-f005:**
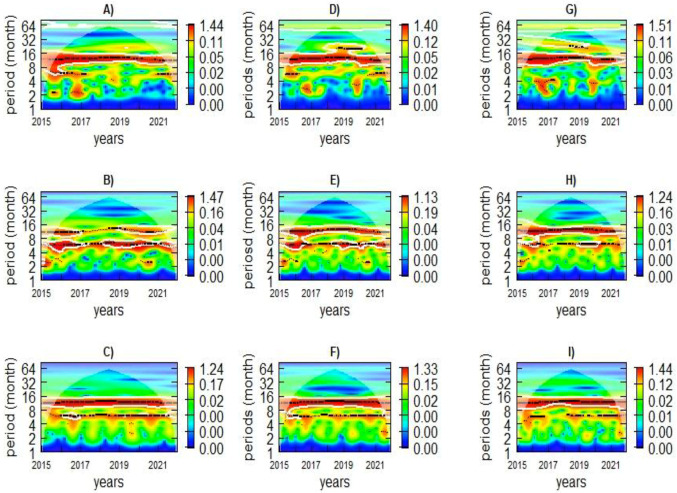
Wavelet power levels of malaria incidence (**A**,**D**,**G**), LST (**B**,**E**,**H**), and rainfall (**C**,**F**,**I**) in the hot/arid (left plots), moderate (middle plots), and cooler/wet zones (right plots), respectively. The cone of influence (COI), where edge effects might influence the analysis, is depicted as a lighter shade. Patterns below the cross-hatched region are considered statistically significant. The color code ranges from blue (low values) to red (high values), indicating increasing significance levels. The white lines outline areas of significance.

**Figure 6 ijerph-21-01487-f006:**
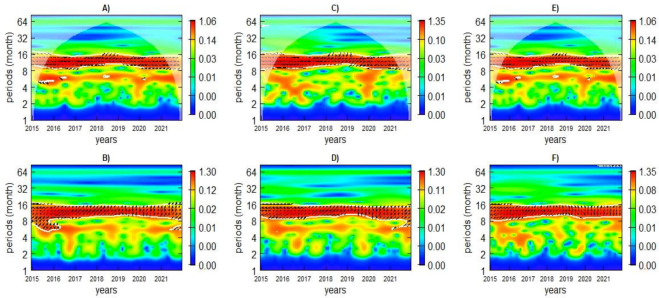
Cross-coherence of LST (**A**,**C**,**E**) and rainfall (**B**,**D**,**F**) with malaria incidence in the hot/arid (left plots), moderate (middle plots), and cooler/wet (right plots) zones, respectively. The arrows indicate the relative phasing.

**Table 1 ijerph-21-01487-t001:** Monthly average malaria cases, climatic factors, and coverage of interventions per region and climatic zones.

Region	LST (°C)	Rainfall (mm)	ACTs	Bednets	Cases per 1000/Month	Climatic Zones
Cascades	27.08	91.89	0.88	0.24	11.00	Cooler/wet
Hauts-Bassins	27.73	90.40	0.91	0.29	12.23	Cooler/wet
Southwest	27.74	91.46	0.89	0.34	17.66	Cooler/wet
Center-North	28.98	58.04	0.88	0.36	7.15	Hot/arid
North	29.07	55.42	0.91	0.29	6.98	Hot/arid
Sahel	29.55	42.37	0.87	0.28	6.87	Hot/arid
Boucle du Mouhoun	28.75	71.12	0.89	0.29	8.92	Moderate
Center	28.63	70.85	0.74	0.21	8.05	Moderate
Center-East	28.83	76.46	0.89	0.27	10.21	Moderate
Center-West	28.38	78.08	0.89	0.33	9.28	Moderate
Center-South	28.33	79.15	0.92	0.33	10.17	Moderate
East	28.73	68.81	0.88	0.22	10.69	Moderate
Plateau-Central	28.85	66.76	0.90	0.43	9.30	Moderate
Overall	28.50	72.37	0.88	0.30	9.92	Overall

**Table 2 ijerph-21-01487-t002:** Posterior inclusion probabilities for various lag times (in months) of LST and rainfall estimated from Bayesian variable stochastic selection (BVSS) for each GARMA model.

Predictor	Lags	GARMA (1, 0)			GARMA (0, 1)			GARMA (1, 1) ^d^		
		Probability of Inclusion			Probability of Inclusion			Probability of Inclusion		
		Cooler/Wet	Moderate	Hot/Arid	Cooler/Wet	Moderate	Hot/Arid	Cooler/Wet	Moderate	Hot/Arid
LST	0	0.05	0.15	0.05	0.05	0.15	0.04	0.05	0.15	0.04
	1	0.40 ^c^	0.35 ^c^	0.2	0.42 ^c^	0.28 ^c^	0.11	0.50 ^c^	0.30 ^c^	0.2
	2	0.3	0.16	0.50 ^c^	0.33	0.18	0.60 ^c^	0.28	0.16	0.51 ^c^
	3	0.1	0.1	0.1	0.1	0.15	0.1	0.17	0.15	0.1
	1	0.06	0.11	0.05	0.05	0.11	0.05	0.05	0.11	0.05
	12	0.05	0.08	0.07	0.03	0.08	0.07	0.03	0.08	0.07
	123	0.04	0.05	0.03	0.02	0.05	0.03	0.02	0.05	0.03
RAINFALL	0	0.05	0.03	0.05	0.05	0.03	0.06	0.05	0.03	0.04
	1	0.1	0.2	0.11	0.1	0.2	0.12	0.1	0.2	0.14
	2	0.34	0.47 ^c^	0.58 ^c^	0.3	0.39 ^c^	0.53 ^c^	0.32	0.44 ^c^	0.50 ^c^
	3	0.38 ^c^	0.13	0.08	0.41 ^c^	0.2	0.1	0.40 ^c^	0.16	0.12
	1	0.05	0.1	0.03	0.06	0.11	0.05	0.05	0.1	0.05
	12	0.05	0.05	0.1	0.05	0.05	0.1	0.05	0.05	0.1
	123	0.03	0.02	0.05	0.03	0.02	0.04	0.03	0.02	0.05
DIC ^a^		11,302	7291	5501 ^b^	11,315	7302	5522 ^b^	11,279	7239	5446 ^b^

^a^ Deviance Information Criterion, ^b^ the lowest DIC, ^c^ corresponds to the highest probability of inclusion, ^d^ the model with the lowest DIC.

**Table 3 ijerph-21-01487-t003:** Posterior estimates (median and 95% BCI) obtained from the GARMA (1, 1) negative binomial models fitted to data from each climatic zone.

Climatic Zones	Parameter	IRR ^a^ (95% BCI)
Cooler/wet	LST_1	0.85 (0.77–0.89) ^b^
	Rain_3	1.36 (1.24–1.48) ^b^
	Bednets	0.81 (0.52–0.95) ^b^
	ACT	1.05 (0.96–1.11)
	AR coefficient	0.61 (0.45–0.77)
	MA coefficient	0.17 (0.10–0.23)
Moderate	LST_1	0.81 (0.74–0.86) ^b^
	Rain_2	1.24 (1.11–1.37) ^b^
	Bednets	0.44 (0.33–0.93) ^b^
	ACT	1.05 (0.95–1.18)
	AR coefficient	0.51 (0.39–0.64)
	MA coefficient	0.35 (0.21–0.55)
Hot/arid	LST_2	0.78 (0.71–0.88) ^b^
	Rain_2	1.51 (1.33–1.70) ^b^
	Bednets ACT	1.08 (0.92–1.14) 0.89 (0.79–0.97) ^b^
	AR coefficient	0.53 (0.41–0.68)
	MA coefficient	0.29 (0.20–0.54)

^a^ Incidence rate ratio, ^b^ Statistically important.

**Table 4 ijerph-21-01487-t004:** Forecast accuracy among the three climatic zones measured by the absolute percentage error (APE) obtained for 1 to 12 months lead times.

Zones	APE for Each Lead Time (Months)												MAPE	RMSE	R^2^
	1	2	3	4	5	6	7	8	9	10	11	12			
Cooler/wet	18.22	15.53	12.28 ^a^	61.92	51.62	30.5	27	26.06	20.19	29.15	23.68	17.8	28	35	0.78
Moderate	24.28	7.35 ^a^	8.76	76.63	110.26	70.55	29.77	16.85	15.27	35.26	88.54	59.76	53	42	0.65
Hot/arid	7.77	3.82 ^a^	6.98	20.87	49.01	98.75	136.65	5.48	22.09	23.25	81.27	78.61	45	39	0.76

^a^ lowest APE, where the highest predictive performance is observed.

**Table 5 ijerph-21-01487-t005:** Summary of interventions/climate factor effect on malaria incidence, lag times, forecasting lead times/accuracy, and seasonal scales by climatic zones.

Characteristic		Climatic Zones	
	Cooler/Wet	Moderate	Hot/Arid
Interventions and climatic factor effect			
Rain	+	+	+
LST	+	+	+
Bednet coverage	+	+	−
ACT coverage	−	−	+
Lag times from the BVSS (in months)			
Rainfall	3	2	2
LST	1	1	2
Forecasting lead times (in months)	3	2	2
Forecasting accuracy			
MAPE (%)	28	53	45
RMSE (%)	35	42	39
Seasonal scale (in months)			
Rainfall	12 (2015–2015)	12 (2015–2021)	12 (2015–2021)
LST	12 (2015–2021)	12 (2015–2021)	12 (2015–2016,2019–2020)
	6 (2015–2016, 2019–2021)	6 (2015–2016, 2019–2021)	6 (2015–2021)
Malaria incidence	12 (2015–2021)	12 (2015–2021)	12 (2015–2021)
			6 (2015–2016)
Phase differences from the wavelets analysis (in months)			
Rainfall over incidence			
12-month period	1.5 (2015–2016)	1.5 (2015–2017)	1.5 (2015–2017)
	2 (2017)	2 (2018–2021)	2 (2018–2021)
	2.5 (2018, 2021)		
	3 (2019–2020)		
LST over incidence			
6-month period	1 (2015–2016),	1.5 (2015–2016),	1.5 (2015–2019),
	1.5 (2020–2021)	1 (2019–2021)	2 (2020–2021)
12-month period	1.5 (2018–2021)	1 (2015–2016),	1.5 (2015),
	1 (2015–2017)	2 (2017–2021)	2 (2016, 2018–2020)

(+) indicates statistically important effect and (−) indicates non-statistically important effect, wavelet assumes non-stationarity, BVSS assumes stationarity.

## Data Availability

The data were extracted from the District Health Information System 2 (DHIS2) of Burkina Faso: https://burkina.dhis2.org/ (accessed on 1 November 2024).

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
