# Peer review of "Impact of Climate Variability and Interventions on Malaria Incidence and Forecasting in Burkina Faso"

_ijerph, 2024, doi:10.3390/ijerph21111487_

Round 1
Reviewer 1 Report
Comments and Suggestions for Authors
The manuscript addresses an important public health issue, focusing on the impact of climate variability and malaria interventions on malaria incidence in Burkina Faso. The study employs advanced statistical techniques, such as Bayesian GARMA models and wavelet analysis, to predict malaria incidence across different climatic zones. While the manuscript has potential, it requires significant revisions to enhance clarity, coherence, and scientific rigor.
Abstract:
1. The abstract is informative but could benefit from more concise language. I suggest removing redundant phrases and focusing on the core findings. Redundant phrases like "Malaria is a major public health burden in Burkina Faso" could be trimmed.Additionally, the abstract should provide a clearer summary of the key statistical results and their implication.
2. Consider adding more specific keywords related to the methodologies used, such as "Bayesian modeling" and "wavelet analysis," to improve the manuscript's visibility.
Introduction:
3. The introduction provides a broad background on malaria and climate interactions but does not clearly articulate the research gap or the novelty of the study. I recommend restructuring this section to highlight the specific problem being addressed and how this study fills a gap in the current literature.
4. The manuscript would benefit from a clearly stated hypothesis or research question to guide the reader through the study.
Methods:
5. While the methods are generally well-described, certain aspects, such as the Bayesian variable stochastic selection (BVSS) and wavelet analysis, require further explanation. Providing a brief rationale for the choice of these methods over others would strengthen the manuscript.
6. Please ensure that all data sources, tools, and software are fully cited and described. This will enhance the reproducibility of the study.
Results:
7. The results section is dense and challenging to navigate. Consider breaking down complex tables and figures into smaller, more digestible components. Additionally, direct references to figures within the text would help guide the reader through the results.
8. The interpretation of the statistical findings could be clearer. For instance, explaining the practical implications of the incidence rate ratios (IRRs) in a more straightforward manner would make the results more accessible to a broader audience.
Discussion:
9.While the discussion compares the findings with existing studies, it lacks depth in explaining why certain results may differ. I recommend exploring potential reasons for these differences and discussing the study's limitations more thoroughly.
10.The manuscript touches on the implications of the findings but could expand on how these results can inform public health policy. Specific recommendations based on the study's findings would be valuable
11. The discussion should include a more robust section on future research directions. Identifying specific unanswered questions or suggesting methodologies for future studies would enhance the manuscript.
Conclusion:
12.The conclusion effectively summarizes the study but could do more to tie the findings back to the research objectives stated in the introduction. Additionally, a discussion of the study's limitations would provide a more balanced conclusion.
13. Consider ending with a strong statement about the importance of adaptive forecasting models in managing malaria under changing climatic conditions.
Comments on the Quality of English Language
The manuscript should maintain a formal and scientific tone throughout. Simplifying overly complex sentences and avoiding colloquial language would improve readability.
Author Response
Comment 1: The abstract is informative but could benefit from more concise language. I suggest removing redundant phrases and focusing on the core findings. Redundant phrases like "Malaria is a major public health burden in Burkina Faso" could be trimmed. Additionally, the abstract should provide a clearer summary of the key statistical results and their implication.
Response: Thank you for your feedback on the abstract. We fully agree that the conciseness and a focus on core findings will improve its clarity. To address this, we revised the abstract (See revised manuscript, lines 11-16; 24-27 and 32-35).
Comment 2: Consider adding more specific keywords related to the methodologies used, such as "Bayesian modeling" and "wavelet analysis," to improve the manuscript's visibility.
Response: Thank you for your suggestion. We added more keywords (See revised manuscript, lines 36-37).
Comment 3: The introduction provides a broad background on malaria and climate interactions but does not clearly articulate the research gap or the novelty of the study. I recommend restructuring this section to highlight the specific problem being addressed and how this study fills a gap in the current literature.
Response: Thank you for your constructive feedback regarding the introduction. We appreciate your insights on the importance of clearly articulating the research gap and the novelty of the study. In response, we have revised the introduction (See revised manuscript, lines 74-117).
Comment 4: The manuscript would benefit from a clearly stated hypothesis or research question to guide the reader through the study.
Response: Thank you for your insightful comment regarding the need for a clearly articulated hypothesis or research question. We agree that this addition will enhance the manuscript's clarity and guide the reader through the study. In response, we have added the research questions at the end of the introduction (See revised manuscript, lines 119-128). The questions explicitly outline our investigation into how climatic factors and malaria interventions interact across different climatic zones in Burkina Faso and how these interactions influence malaria incidence.
Comment 5: While the methods are generally well-described, certain aspects, such as the Bayesian variable stochastic selection (BVSS) and wavelet analysis, require further explanation. Providing a brief rationale for the choice of these methods over others would strengthen the manuscript.
Response: Thank you for this valuable suggestion. We have expanded the methodology section to provide an additional explanation of both Bayesian variable stochastic selection (BVSS) and wavelet analysis. Specifically:
Bayesian Variable Stochastic Selection (BVSS): We have clarified that BVSS was selected due to its ability to efficiently handle uncertainty in predictor selection, particularly in complex models involving multiple climatic and intervention variables. BVSS enables a rigorous approach to determining the most relevant lag times and climatic factors affecting malaria incidence, which is especially beneficial for our study given the variability across the three climatic zones (See revised manuscript, lines 186-190).
Wavelet Analysis: We have included a rationale for using wavelet analysis, emphasizing that it is a powerful tool for identifying periodic patterns and fluctuations across multiple time scales. Given malaria's seasonal dynamics, wavelet analysis provides critical insights into how these dynamics vary over time and across different climatic zones, which cannot be captured with traditional time series methods (See revised manuscript, lines 211-216).
Comment 6: Please ensure that all data sources, tools, and software are fully cited and described. This will enhance the reproducibility of the study.
Response: Thank you for highlighting the importance of reproducibility in research. We have carefully reviewed the manuscript and have ensured that all data sources, tools, and software utilized in the study are fully cited and described. This includes specifying the datasets used for analysis, any software applications for statistical modeling, and relevant libraries or packages (See revised manuscript, method section).
Comment 7: The results section is dense and challenging to navigate. Consider breaking down complex tables and figures into smaller, more digestible components. Additionally, direct references to figures within the text would help guide the reader through the results.
Response: Many thanks for the suggestion to consider breaking down the tables and figures to enhance clarity. Given the study’s focus on comparing results across the three climatic zones, it is essential to maintain unified tables and figures where feasible. This approach enables straightforward comparisons across zones and enhances the interpretability of differences related to predictors, model probabilities, incidence rates, and forecasting errors. However, we made several layout enhancements to improve readability. For instance, borders have been added to each table to distinguish rows and columns clearly, making the information easier to follow and enhancing visual organization. Results are now grouped by climatic zone wherever possible. This organization highlights regional differences and allows for quick comparison across zones. Furthermore, to draw attention to critical values, I have used bold formatting and highlighted significant results within the tables, enabling readers to focus on key findings immediately (See revised manuscript, results section from Table 2 to Table 5).
Comment 8: The interpretation of the statistical findings could be clearer. For instance, explaining the practical implications of the incidence rate ratios (IRRs) in a more straightforward manner would make the results more accessible to a broader audience.
Response: Thank you for your suggestion to improve the interpretation of our statistical findings. We have included a statement describing the practical implication of IRRs in the methods (See revised manuscript, lines 207-209) and we also revised the results section (See revised manuscript, lines 283-295).
Comment 9: While the discussion compares the findings with existing studies, it lacks depth in explaining why certain results may differ. I recommend exploring potential reasons for these differences and discussing the study's limitations more thoroughly.
Response: Thank you for your valuable feedback regarding the depth of our discussion. We appreciate your suggestion to explore potential reasons for differences in our findings compared to existing studies and to discuss the limitations of our research. We revised the discussion, and we provided limitations of our study (See revised manuscript, lines 496-510).
Comment 10: The manuscript touches on the implications of the findings but could expand on how these results can inform public health policy. Specific recommendations based on the study's findings would be valuable
Response: Thank you for the suggestion to expand on the policy implications of our findings. In response, we have enhanced the Discussion section to outline specific recommendations for public health policy, which can be directly informed by our study’s results (See revised manuscript, lines 512-524).
Comment 11: The discussion should include a more robust section on future research directions. Identifying specific unanswered questions or suggesting methodologies for future studies would enhance the manuscript.
Response: Thank you for your feedback regarding the inclusion of future research directions. We have added a section at the end of the discussion that suggests future studies. (See revised manuscript, lines 524-537).
Comment 12: The conclusion effectively summarizes the study but could do more to tie the findings back to the research objectives stated in the introduction. Additionally, a discussion of the study's limitations would provide a more balanced conclusion.
Response: Thank you for your valuable feedback on the conclusion. In response to your suggestion, we have revised the conclusion to better align with the research objectives (See revised manuscript, lines 557-560).
Comment 13: Consider ending with a strong statement about the importance of adaptive forecasting models in managing malaria under changing climatic conditions.
Response: We appreciate your recommendation for a stronger closing statement regarding the importance of adaptive forecasting models. We have incorporated a concluding remark that highlights the critical role these models play in managing malaria under changing climatic conditions (See revised manuscript, lines 560-566).
Comment 14: The manuscript should maintain a formal and scientific tone throughout. Simplifying overly complex sentences and avoiding colloquial language would improve readability.
Response: Thank you for your valuable feedback regarding the manuscript's readability. We appreciate your suggestions to maintain a formal and scientific tone throughout. In response, we have thoroughly made language revisions. This includes rephrasing sections for clarity and conciseness, ensuring that the language used is appropriate for a scientific audience.
Reviewer 2 Report
Comments and Suggestions for Authors
After a thorough evaluation of the manuscript, I find that it well-written and presented in a clear and comprehensible manner. The authors have demonstrated a commendable adherence to scientific rigor throughout the entirety of their research process. The introduction offered a solid rationale for the study, effectively establishing the relevance and importance of the research within the broader scientific context. The methodology section is adequately detailed, providing sufficient clarity on the experimental design and procedures, while the results are presented in a clear and concise manner. Furthermore, the discussion effectively situates the findings within the existing literature, providing a well-rounded analysis of their significance and potential implications for public health and meteorological studies. The statistical methods employed are robust, and the data analysis is thorough, thereby ensuring that the conclusions drawn are firmly supported by the evidence provided in the study.
Minor Comment: One potential limitation pertains to the use of malaria case data. Specifically, there are concerns regarding the spatial validity of the case data, as individuals seeking healthcare may not necessarily reside in the same region where the data is aggregated. As a result, they may be exposed to different climatic or environmental conditions than those associated with the region under which they are categorized. This discrepancy could introduce challenges in accurately relating malaria case data to climate variables. The authors should provide additional clarification on how the malaria case data from Burkina Faso's district health system was processed to address this issue, particularly with regard to filtering or adjusting the data to account for potential spatial mismatches.
Author Response
General comment: After a thorough evaluation of the manuscript, I find that it well-written and presented in a clear and comprehensible manner. The authors have demonstrated a commendable adherence to scientific rigor throughout the entirety of their research process. The introduction offered a solid rationale for the study, effectively establishing the relevance and importance of the research within the broader scientific context. The methodology section is adequately detailed, providing sufficient clarity on the experimental design and procedures, while the results are presented in a clear and concise manner. Furthermore, the discussion effectively situates the findings within the existing literature, providing a well-rounded analysis of their significance and potential implications for public health and meteorological studies. The statistical methods employed are robust, and the data analysis is thorough, thereby ensuring that the conclusions drawn are firmly supported by the evidence provided in the study.
Response: Many thanks to Reviewer #2 for the positive feedback.
Comment 1: One potential limitation pertains to the use of malaria case data. Specifically, there are concerns regarding the spatial validity of the case data, as individuals seeking healthcare may not necessarily reside in the same region where the data is aggregated. As a result, they may be exposed to different climatic or environmental conditions than those associated with the region under which they are categorized. This discrepancy could introduce challenges in accurately relating malaria case data to climate variables. The authors should provide additional clarification on how the malaria case data from Burkina Faso's district health system was processed to address this issue, particularly with regard to filtering or adjusting the data to account for potential spatial mismatches.
Response: Thank you for highlighting this important point regarding the spatial validity of malaria case data. We fully agree that patients seeking health care may sometimes come from neighboring regions and may experience different environmental and climatic conditions than those in the area where their cases are recorded. However, the available data did not allow for direct filtering to account for individual movement across district boundaries, we recognized this limitation and its potential impact on accurately linking case data to local climatic factors. We have included this as one of the study limitations in the revised manuscript (See lines 498-502). In addition, we acknowledge that further studies incorporating spatial tracking or patient origin data could help refine these associations in future analyses.
Reviewer 3 Report
Comments and Suggestions for Authors
Dear Author(s)
Although your manuscript is an informative and useful text, one comment has been arisen that as we know RDT technique has some advantages and some disadvantages. One of the disadvantages is misdiagnosing when the parasitaemia is less than 10 parasites per microliter of blood. How did you check such mater to make sure the correctness of the diagnosing results?
Author Response
Comment 1: Although your manuscript is an informative and useful text, one comment has been arisen that as we know RDT technique has some advantages and some disadvantages. One of the disadvantages is misdiagnosing when the parasitaemia is less than 10 parasites per microliter of blood. How did you check such mater to make sure the correctness of the diagnosing results?
Response: We thank reviewer #3 for bringing attention to this important aspect of malaria diagnosis. In our study, we used monthly aggregated data at the regional level from Burkina Faso's District Health Information System (DHIS2), which includes confirmed malaria cases diagnosed by both microscopy and RDTs. While it is true that individual misdiagnoses can occur with RDTs - especially when parasitaemia is below detectable levels - the aggregated nature of our data reduces the likelihood that these misdiagnoses will significantly affect the overall trends and patterns we observe. By analyzing the data at this broader level, we capture the more stable trends in malaria incidence that are less susceptible to fluctuations caused by individual diagnostic errors.